

# The effect of combined dietary supplementation of herbal additives on carcass traits, meat quality, immunity and cecal microbiota composition in Hungarian white geese

Guilin Fu[1,*], Yuxuan Zhou[1,*], Yupu Song[1], Chang Liu[2], Manjie Hu[1], Qiuyu Xie[1], Jingbo Wang[1], Yuxin Zhang[1], Yumeng Shi[1], Shuhao Chen[1], Jingtao Hu[1] and Yongfeng Sun[1]

[1] College of Animal Science and Technology, Jilin Agricultural University, Changchun, China
[2] Changchun Animal Husbandry Service, Changchun, China
[*] These authors contributed equally to this work.

Corresponding authors
Jingtao Hu, jingtaoh@jlau.edu.cn
Yongfeng Sun,
sunyongfeng@jlau.edu.cn

## ABSTRACT

The present study was performed to investigate the effects of dietary supplementation with herbal additives on meat quality, slaughter performance and the cecal microbial community in Hungarian white geese. A total of 60 newborn geese were assigned equally into the control group (CON) and the herbal complex supplemented group (HS). The dietary supplementations consisted of Compound Herbal Additive A (CHAA) including *Pulsatilla, Gentian* and *Rhizoma coptidis*, and Compound Herbal Additive B (CHAB) containing *Codonopsis pilosula, Atractylodes, Poria cocos* and *Licorice*. The geese in the HS group received a basal diet supplemented with 0.2% CHAA from day 0 to day 42 at the postnatal stage. Then from day 43 to day 70, the geese in HS group were provide a basal diet with 0.15% CHAB. The geese in the CON group were only provided with the basal diet. The results showed that the slaughter rate (SR), half chamber rates (HCR), eviscerated rate (ER) and breast muscle rate (BMR) in the HS group tended to increase slightly compared with the CON group (ns). In addition, the shear force, filtration rate and pH value of breast muscle and thigh muscle in the HS group were slightly enhanced compared to the CON group (ns). Significant increased levels in carbohydrate content, fat content and energy ($P < 0.01$) and significant decreased levels in cholesterol content ($P < 0.01$) were observed in the muscle of the HS group. The total amino acid (Glu, Lys, Thr and Asp) content in the muscle increased in HS group than in the CON group ($P < 0.01$). Dietary herb supplementations significantly increased the levels of IgG in serum ($P < 0.05$) on day 43 and higher levels of IgM, IgA and IgG ($P < 0.01$) were also observed in the HS group on day 70. Furthermore, 16S rRNA sequencing results indicated that herbal additives increased the growth of beneficial bacteria and inhibited the proliferation of harmful bacteria in the geese caecum. Altogether, these results offer crucial insights into the potential benefits of incorporating CHAA and CHAB into the diets of Hungarian white goose. The findings indicate that such supplementations could significantly improve meat quality, regulate the immune system and shape the intestinal microbiota composition.

# INTRODUCTION

To produce qualified livestock or poultry whose meat are low in fat and high in proteins is highly required to fulfill people's growing need in recent years. (*Li et al., 2019*). Goose meat is considered as a healthy food with high content in amino acids and minerals exerting beneficial effects on human health (*Simopoulos, 2008*). Several diseases, including cardiovascular problems, inflammatory and autoimmune disorders, are associated with the imbalance intake of n-6 and n-3 polyunsaturated fatty acids in the diet (*Qi et al., 2010*). The high concentration of polyunsaturated fatty acids in goose flesh makes it a healthier choice for consumers. For the moment, goose production in China has accounted for approximately 94% of the gross production of global industry (*Liu & Zhou, 2013*; *Yu et al., 2020*). In the past few decades, antibiotic growth promoters (AGPs) have been extensively applied in poultry diets to prevent and treat disease and improve goose growth, as antibiotic feed utilization could maximize profits and efficiency. However, the abuse of antibiotics has led to an increasing number of bacterial drug-resistant strains in animal products, posing a serious threat to human and animal health. Therefore, AGPs were banned by the European Union in 2006 (*Villanueva, 2012*), North America in 2017, and China in 2020, successively. It is urgent to develop safe and effective additives to serve as AGPs alternatives in the animal production industry.

A growing body of research has been conducted on Chinese herbal medicines and their purified components and indicated they could serve as novel growth enhancers and antibiotic alternatives (*Seidavi et al., 2021*). Chinese herbal medicine could improve immunity, reduce inflammation, provide antibacterial and antioxidant properties (*Hernandez et al., 2004*; *Acamovic & Brooker, 2005*; *Giannenas et al., 2018*). Consequently, they could protect the intestinal mucosal structure in poultry and affect the fowl gut microbiota (*Abd El-Hack et al., 2022*; *Khalaji et al., 2011*; *Abolfathi et al., 2019*). *Pulsatilla, Gentian, Rhizoma coptidis, Codonopsis pilosula, Atractylodes, Poria cocos* and *Licorice* were selected with the such properties and were utilized in this research (*Zhong et al., 2022*; *Mirzaee et al., 2017*; *Wang et al., 2019*; *Fu et al., 2018*; *Bailly, 2021b*; *Tian et al., 2019*).

The large-scale production and intensification of animal husbandry could shorten the feeding cycle, at the same time generate severe metabolic burden, leading to the decline of meat quality (*Xing et al., 2019*). Chinese herbal medicine exerted antioxidant activity and potential beneficial effects on poultry meat quality and immunity (*Qaid et al., 2022*; *Xie et al., 2022*). Gentian, as a bittering agent, could affect digestion by increasing appetite in rats, stimulating bitter receptor cells in gastrointestinal tracts and thus improving nutrient absorption and muscle water retention (*Mirzaee et al., 2017*). *Licorice* could improve the meat quality both in broilers and fattening pigs. Flavonoids and triterpene saponins are the main active substances of *licorice* (*Ahmed et al., 2016*; *Qiao et al., 2022*) *Licorice* extract is known to improve immune response by increasing interferon and globulin

levels in animals (*Toson et al., 2022*). Moreover, *Poria cocos* polysaccharide has also been demonstrated to perform multiple immune effects by promoting antibody production in B lymphocytes and the spleen, increasing serum IgG levels, and enhancing the phagocytosis of macrophages (*Tian et al., 2019*; *Pu et al., 2019*). Polysaccharides from *Codonopsis pilosula* have been reported to protect the intestinal mucosal immune barrier, to maintain intestinal homeostasis (*Li et al., 2022*).

Previous studies indicated that *Pulsatilla*, *Rhizoma coptidis*, *Codonopsis pilosula* and *Atractylodes* could regulate gut microbiota composition (*Li et al., 2020*; *Wang et al., 2019*; *Bailly, 2021a*; *Bailly, 2021b*). *Pulsatilla* chinensis saponins (PRS) are the main active component of *Pulsatilla* and its antioxidant and immunomodulatory functions have been extensively studied (*Li et al., 2020*). In a rat model of dextran sodium sulfate (DSS)-induced ulcerative colitis (UC), the administration of PRS regulated the composition and biodiversity of the gut microbiota, significantly improving UC symptoms and reducing inflammation (*Liu et al., 2021*). In addition, *Rhizoma coptidis* has various pharmacological effects containing alkaloids such as berberine, coptisine and palmatine, which could exert antioxidant, and anti-viral effects (*Lyu et al., 2021*; *Zhang et al., 2021*; *Wang et al., 2018*). Studies have shown that *Codonopsis* polysaccharides could enhance intestinal mucosal immune function by stimulating the secretion of sIgA (*Fu et al., 2018*; *Zou et al., 2019*). According to previous studies, sesquiterpene lactams and lactones are the main active components of *Atractylodes macrocephala*, which perform excellent anti-inflammatory and antioxidant activities (*Bailly, 2021b*). *Atractylodes* are commonly used to heal gastrointestinal disorders due to their potential role in regulating intestinal microbiota (*Feng et al., 2020*; *Wang et al., 2022*). Previous studies have indicated that several herbal medicines are usually combined to achieve the synergetic effect better than single herb based on their similar therapeutic properties (*Xu et al., 2022*; *Zhou et al., 2016*; *Li et al., 2021*). In our study, CHAA (*Pulsatilla, Gentian* and *Rhizoma coptidis*) and CHAB (*Codonopsis pilosula, Atractylodes, Poria cocos* and *Licorice*) were prepared respectively. However, most research on herbal supplementations has been performed in mammals rather than in poultry (*Li et al., 2016*; *Huang et al., 2021*; *Li et al., 2021*). Therefore, this study aims to investigate the effects of CHAA and CHAB on carcass traits, meat quality, immunity standard and intestinal microbiome in Hungarian white goose.

## MATERIALS & METHODS

### Schematic overview of the experimental program

CON group: geese were fed with the basal diet for 70 days. HS groups: geese were fed with the basal diet supplemented with CHAA (day 0-42) and CHAB (day 43-70). On day 42 and 70, serum was collected to determine serum IgA, IgM, IgG, and the contents of the cecum were collected for 16S sequencing. On day 70, carcass traits and meat quality were detected after slaughter experiments (Fig. 1).

### Ethics Statement

All the experiment animals were housed and raised according to the guidelines set by the Institute of Animal Care and Use Committee of Jilin Agricultural University (Approval

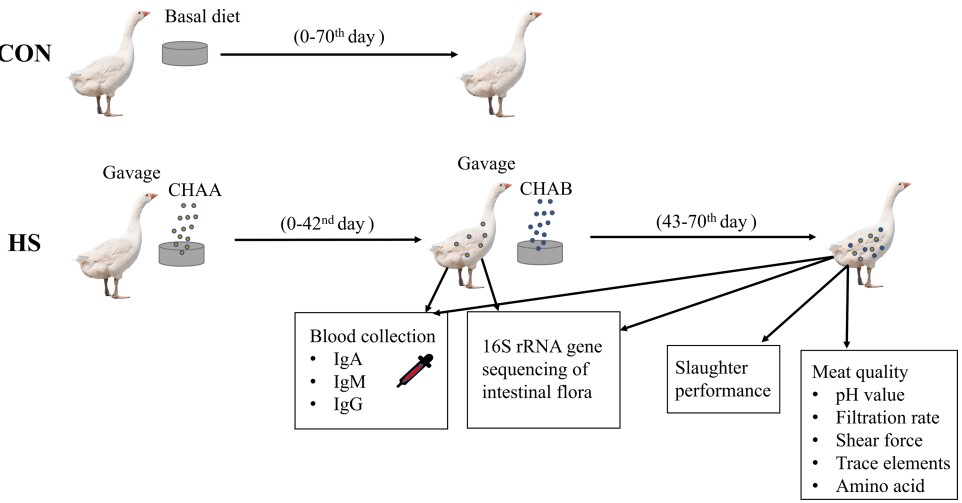

**Figure 1** **Schematic overview of the experimental program.** Experimental Hungarian white geese grouping, treatment and specific experiments.

number: No. 2020 04 30 001, 12th April 2020). All the geese were maintained in pathogen-free conditions and treated under the International Association for Assessment and Accreditation of Laboratory Animal Care policies and certification.

### Animals

A total of 60 one-day old Hungarian white geese were involved in this research. The geese were purchased from the breeding base of the Goose Research Center of Jilin Agricultural University and the average weights of male and female goslings were about 4300 g and 3600 g, respectively.

### Preparation and characterization of Chinese herbal complex

The CHAA and CHAB were prepared by Changchun General Animal Husbandry Station. The preparation process contains drying, crushing and pulverizing Chinese herbs into a fine powder, then passed through an 800-mesh sieve for uniformity. The powdered herbs were mixed in proportion to create an herbal compound, then stored at room temperature (25 °C). The method was described by (*Xu et al., 2022*). Under the ratio value of 2:1:1, *Pulsatilla*, *Gentian* and *Rhizoma coptidis* were compound to produce CHAA, and *Codonopsis pilosula*, *Atractylodes*, *Poria cocos* and *Licorice* under the ratio of 3:3:3:2 was used to produce CHAB. Saponin in herbal mixtures are primarily derived from *Pulsatilla*, *Codonopsis pilosula*, and *Licorice*, and alkaloids are the main active substances in *Rhizoma coptidis* and *Atractylodes*. Total flavonoids are derived from *Rhizoma coptidis* and *Licorice*, while total polysaccharides are derived primarily from *Codonopsis pilosula*, *Atractylodes*, and *Poria cocos* (*Fu et al., 2018*; *Li et al., 2022*; *Wu et al., 2021*). Polysaccharides, flavonoids, saponins and alkaloids are the main antioxidant compounds (*Xu et al., 2022*).

**Table 1   Ingredients and composition of basal diets (DM basis) %.**

| Items | Starter 0 to 42 d | Grower 43 to 70 d |
|---|---|---|
| Ingredients, % | | |
| Corn | 52.00 | 53.00 |
| Soybean meal | 22.00 | 14.00 |
| Wheat bran | 8.00 | 15.00 |
| Fish meal | 4.00 | 1.00 |
| Corn gluten meal | 4.00 | 2.00 |
| Stone meal | 6.00 | 6.00 |
| Calcium hydroxide | 1.50 | 1.80 |
| Soybean oil | 1.20 | 1.50 |
| Salt | 0.30 | 0.30 |
| Rice bran | 0.00 | 4.00 |
| Additives | 1.00 | 1.00 |
| Total | 100 | 100 |
| Chemical composition, % | | |
| CP | 19.68 | 14.82 |
| CF | 3.00 | 6.88 |
| MET | 0.67 | 0.55 |
| LYS | 1.34 | 1,03 |
| THR | 0.87 | 0.66 |
| Ca | 0.65 | 0.59 |
| P | 0.48 | 0.41 |
| ME, kcal/kg | 13.02 | 12.75 |

**Notes.**

ME, Metabolizable energy; CP, crude protein; CF, crude fiber; Ca, calcium; P, phosphorus; LYS, lysine; Met, Methionine.

### Experimental design

The geese were randomly divided into two groups of 30 animals per group. The geese in the CON group were fed with the basal diet (corn–soybean) during the entire experiment (Table 1), and the geese in the HS group were fed with a basal diet (corn–soybean) adding 0.2% of CHAA on day 0-42 and 0.15% of CHAB on day 43-70. The geese were fed twice a day. From day 7 to 15, the ventilation system was activated for 5-7 h per day and the working time was reduced on rainy days. The ambient temperature was maintained at 30 $\pm 1$ °C during the first week, gradually decreased to 24 $\pm 1$ °C in the second week and then exposed to natural environmental conditions. The artificial lighting program was 23 $\pm 1$ h of light followed by 1 h of darkness until day 10, 18 $\pm 1$ h from day 11 to day 13 and natural light was provided until day 70.

### Serum antibodies detection

Blood samples were collected from the wing vein of geese in each group on day 42 and 70. The serum samples were obtained by centrifuging blood cells at 3,000 g for 10 min (4 ° C) and then stored at −20 °C for further detection. Concentrations of immunoglobulin M (IgM), immunoglobulin G (IgG) and immunoglobulin A (IgA) were determined utilizing

ELISA kits (Nanjing Ao Qing Biotechnology Co. Ltd., Jiangsu, China) following the manufacturer's instructions. Each measurement was replicated three times.

### Carcass characteristics

On day 70, 10 geese from each group were randomly selected to compute all carcass parameters, weighted, slaughtered through complete bleeding. The slaughter process was conducted in a commercial slaughterhouse following the standard procedures (*Miao et al., 2020*). The liver, heart, gizzard, neck and abdominal fat were excised and weighed to determine the carcass weight (CW, g). The slaughter weight (SW, g) was recorded and the slaughter rate (SR) was calculated according to the formula below:

$$SR(\%) = \frac{CW}{SW} X 100\%$$

The breast muscle and thigh muscle were recorded separately. The half chamber rate (HCR) and the eviscerated rate (ER) were calculated by the following equations:

HCR (%) = Half chamber weight/Live weight ×100%. ER (%) = Eviscerated weight/Live weight ×100%. The ratio of breast and thigh muscle was calculated using the percentage of total chamber weight.

### Meat quality indicators

Breast and thigh meat samples were used for pH value, shear force and filtration rate analysis to measure the quality of the meat. The pH value was evaluated using a microprocessor pH meter (ATF-500, Japan Kyoto Electronics Co., Ltd. Japan). The shear force (g) was measured using a digital meat tenderness meter (J C-LM3; Matthaus, Neu-Isenburg, Germany) and the filtration rate was evaluated using the procedures in previous description (*Miao et al., 2020*). Moreover, a pressure gravimetric analysis was used to determine the filtration rate, applying the following equation:

Filtration rate(%) = (initial weight − final weight)/initial weight × 100%.

## Muscle chemical composition detection

Chemical composition of muscle was analyzed according to the following standards. The contents of moisture, protein, fat, carbohydrates, energy and cholesterol were determined according to the corresponding methods of the Chinese National Food Safety Standard (GB 5009.3-2016) (GB 5009.5-2016) (GB 5009.6-2016) (GB 28050-2011) (GB 28050-2011) (GB5009.128-2016), respectively. Additionally, the contents of trace elements (Zn, Fe, Ca, P, Na and Se) were also measured following GB 5009. 14-2017, GB 5009. 90-2016, GB 5009. 92-2016 GB 5009.87-2016, GB 5009.91-2017 and GB 5009.93-2017, individually.

## Amino acid composition analysis

Amino acid levels (Asp, Thr, Ser, Glu, Gly, Ala, Val, Met, Leu, Ile, Tyr, Phe, His, Lys, Arg and Pro) were measured according to the procedures GB5009.124-2016. The amino acid content in muscle was performed on an L-8900 automatic amino acid analyzer (Hitachi, Japan).

### DNA Extraction, 16S rRNA sequencing and bioinformatic analysis

Caecum contents were aseptically collected from geese and stored at −80 °C before DNA extraction and sequencing. Total DNA was extracted using the Hi Pure Soil DNA Kits (Ovison, Beijing, China) following the manufacturer's instructions. The V3-V4 hypervariable region of the bacterial 16S rRNA gene was amplified by PCR. The forward primer 341F (CCTACGGGNGGCWGCAG) and the reverse primer 806R (GGACTACHVGGGTATCTAAT) were used. The PCR products were confirmed with 2% agarose gel electrophoresis, purified with the DNA Gel Extraction Kit (Axygen Biosciences, Union City, CA, USA) and sequencing libraries were generated using the SMR T bell TM Template Prep Kit (PacBio, Menlo Park, CA, USA). Library quality was assessed and sequenced on an Illumina MiSeq platform.

FLASH was used to merge raw sequencing data into tags (version 1.20). Then paired sequences were merged and divided into the Operational Taxonomic Units (OTUs) utilizing Usearch (version 2.7.1) and representative sequences were clustered at 97% sequence similarity. Subsequently, all representative sequences were compared using MUSCLE (version 3.8.31) software to construct the phylogenetic relationship, and further, the data of all samples were normalized to perform the alpha and beta diversity analyses. The final weighted and weighted UniFrac distances were calculated using QIIME (version 1.7.0), to compare microbial structures between different samples. In addition, the samples were clustered based on an unweighted or weighted UniFrac distance matrix using the unweighted pair group method (UPGMA) performed in QIIME (version 1.7.0). Moreover, linear discriminant analysis (LDA) and effect size (LEfSe) software were used to identify the significant differences in microorganisms between groups. Finally, to illustrate differences in microbial composition, LDA scores were calculated and taxonomic cladograms of microbial composition were generated.

### Statistical analysis

All the results on growth performance, meat quality, and serum globulin concentration were analyzed using the $T$-test of the SPSS software (version 20) and statistics information were recorded with WPS2020. The diagrams were constructed by Graph Prism (version 5.0) software. All data expressed as the mean ±standard error (SE). Statistical significance is indicated as ns, no significance, $*p < 0.05$, $**p < 0.01$ and $***p < 0.001$.

## RESULTS

### Serum IgM, IgG and IgA concentrations

The effects of the herbal mixtures on IgG, IgA and IgM in geese serum were shown in Fig. 2. The results indicated that adding CHAA and CHAB in basal diet had no effect on levels of serum IgA and IgM on day 43 (ns), and promoted significantly the secretion levels of IgG ($P = 0.0027$, $F = 1.088$, R-sq $=0.4027$). On day 70, there were a significant increased concentration of serum IgA ($P < 0.0001$, $F = 7.443$, R-sq $=0.9245$), IgM ($P < 0.0001$, $F = 2.204$, R-sq $=0.6930$) and IgG ($P = 0.0005$, $F = 4.322$, R-sq $=0.5032$) in the HS group than CON group ($P < 0.01$).

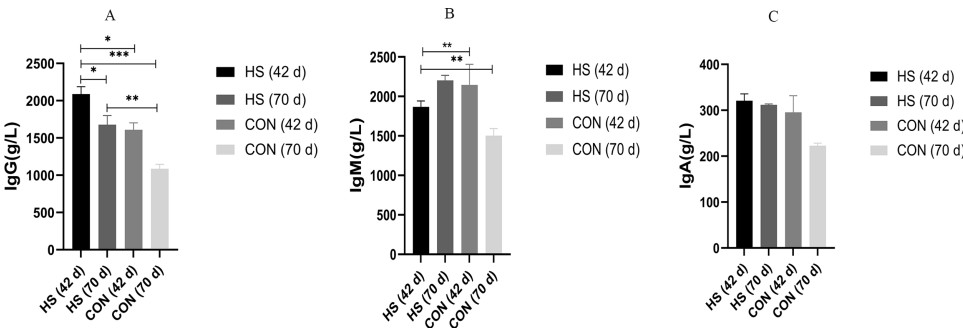

**Figure 2** **Effects of herbal additives on serum Ig parameters in Hungarian white geese at different ages.**
HS = herbal complex supplement group. Addition of CHAA during the initial stage (from days 0 to 42) of
geese feeding (HS 42d). Addition of CHAB during the growth stage (from days 43 to 70) of geese feeding
(HS 70d). *Significant difference compared to the Control group (*$P < 0.05$, **$P < 0.01$, ***$P < 0.001$,
****$P < 0.0001$). All data was presented with appropriate standard error, as mean values ± SEM.

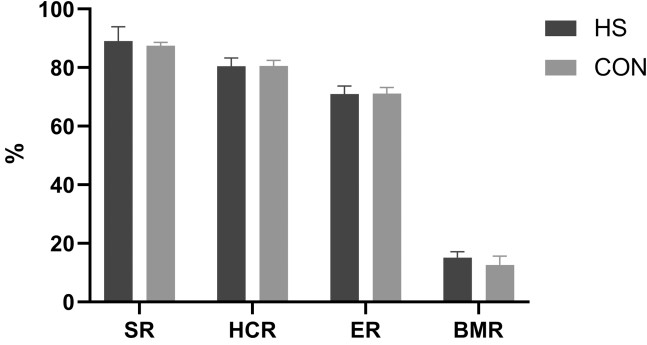

**Figure 3** **Effects of herbal mixture on carcase traits of Hungarian white geese.** SR, slaughter rate; HCR,
half carcass rate; ER, eviscerated rate; BMR, breast t muscle rate.

## Carcass traits

The impact of CHAA and CHAB powders on the carcass traits was shown in Fig. 3. The SR,
HCR, ER and BMR indexes in the HS group were slightly higher than those in the CON
group on day 70 respectively ($P > 0.05$).

## Meat quality

Characteristics of the meat quality (pH, filtration rate and shear force) from the breast and
thigh muscle samples were presented in Table 2. No significant differences were observed
between the CON group and the HS group.

## Muscle chemical composition

The chemical compositions of the meat were detected and the results were shown in Fig. 4.
There was no significant effect of dietary treatment on the moisture or protein levels
between the two groups. Besides, significant increased levels in the carbohydrate content
($P < 0.0001$, $F = 2.333$, R-sq $= 0.9894$), fat content ($P = 0.0009$, $F = 9.333$, R-sq $= 0.9509$)
and energy ($P = 0.0005$, $F = 2.154$, R-sq $= 0.9626$) were observed and significant decreased

**Table 2  Effects of Herbs supplementation on meat quality of geese.**

| muscle | Group | Shear force (g) | Filtration rate (%) | PH value |
|---|---|---|---|---|
| Breast muscles | HS | 102.19 ± 11.13[a] | 24.21 ± 9.6[a] | 5.58 ± 0.05[a] |
| | Control | 82.00 ± 10.83[a] | 24.00 ± 7.8[a] | 5.50 ± 0.02[a] |
| Thigh muscles | HS | 71.39 ± 25.69[a] | 19.64 ± 5.92[a] | 6.22 ± 0.12[a] |
| | Control | 59.70 ± 13.91[a] | 16.08 ± 6.43[a] | 6.05 ± 0.10[a] |

Notes.
[a–c]Means within a row with different letters differ significantly ($P < 0.05$). Data are presented as the mean ± SEM.

level in cholesterol content ($P < 0.0001$, $F = 1.105$, R-sq $=0.9933$) were observed in the HS group than the CON group (Fig. 4A). For the contents of Zn ($P = 0.0005$, $F = 3.083$, R-sq $=0.9626$), P ($P = 0.0124$, $F = 12.18$, R-sq $=0.9626$), Fe ($P < 0.0001$, $F = 12$, R-sq $=0.9999$), Na ($P < 0.0001$, $F = 1.333$, R-sq $=0.9978$), Ca ($P < 0.0001$, $F = 33.33$, R-sq $=0.9880$) and Se($P < 0.0001$, $F = 204.8$, R-sq $=0.9984$) (Fig. 4B) in the muscle, significant decreased levels were found in the HS group in comparison to the CON group.

### Effect of herbal mixture on amino acid composition

The amino acid levels in muscle were shown in Fig. 4C. Results indicated that the total amino acid ($P < 0.0001$, $F = 4.823$, R-sq $=0.9931$), amount of Glu ($P < 0.0001$, $F = 1.750$, R-sq $=0.9928$), Lys ($P = 0.0009$, $F = 3.250$, R-sq $=0.9511$), Thr ($P = 0.0092$, $F = 3.583$, R-sq $=0.8478$), Arg ($P = 0.0033$, $F = 2.154$, R-sq $=0.9070$) and Asp ($P = 0.0029$, $F = 2.926$, R-sq $=0.9137$) in the muscle of the HS group showed significant increased levels ($P < 0.01$) compared with the CON group respectively. On the other hand, the contents of Ile ($P = 0.0319$, $F = 5.895$, R-sq $=0.7232$) in the muscle samples of the HS group were higher than those of the CON group.

### Gut microbial diversity and composition

As shown in Fig. 5A, we found that the Shannon sparsity curve was relatively flat. In addition, the rank abundance curves (Fig. 5B) showed that the abundance difference of OTUs in the community was small enough. There were no significant differences in the Chao1 index, the observed-species index, or the Simpson index between the HS and CON groups (Fig. 5D). Principal coordinates analysis (PCoA) showed that the microbial structure of the HS and CON groups was similar on days 43 and 70, separately. Moreover, the cecal microbial community structure in the HS group on day 43 were significantly different from the HS group on day 70 (Fig. 5C).

### Microbial taxa analysis

The study analyzed the relative abundance of the two groups at the phylum and genus levels respectively. At the phylum level, *Firmicutes*, *Proteobacteria* and *Bacteroidetes* were the dominant phyla in the HS group. The cecal microbiota of the geese was dominated by the *Firmicutes* (52.46%), *Proteobacteria* (25.61%) and *Bacteroidetes* (5.47%) in the HS group on day 70 (Fig. 6A). At the genus level, the taxonomic unit with the highest abundance was Lactobacillus (5.73%) in the HS group on day 70. What is more, the CON group had a significantly higher abundance of *Escherichia-Shigella* (30.89%), *Desulfovibrio* (5.37%)

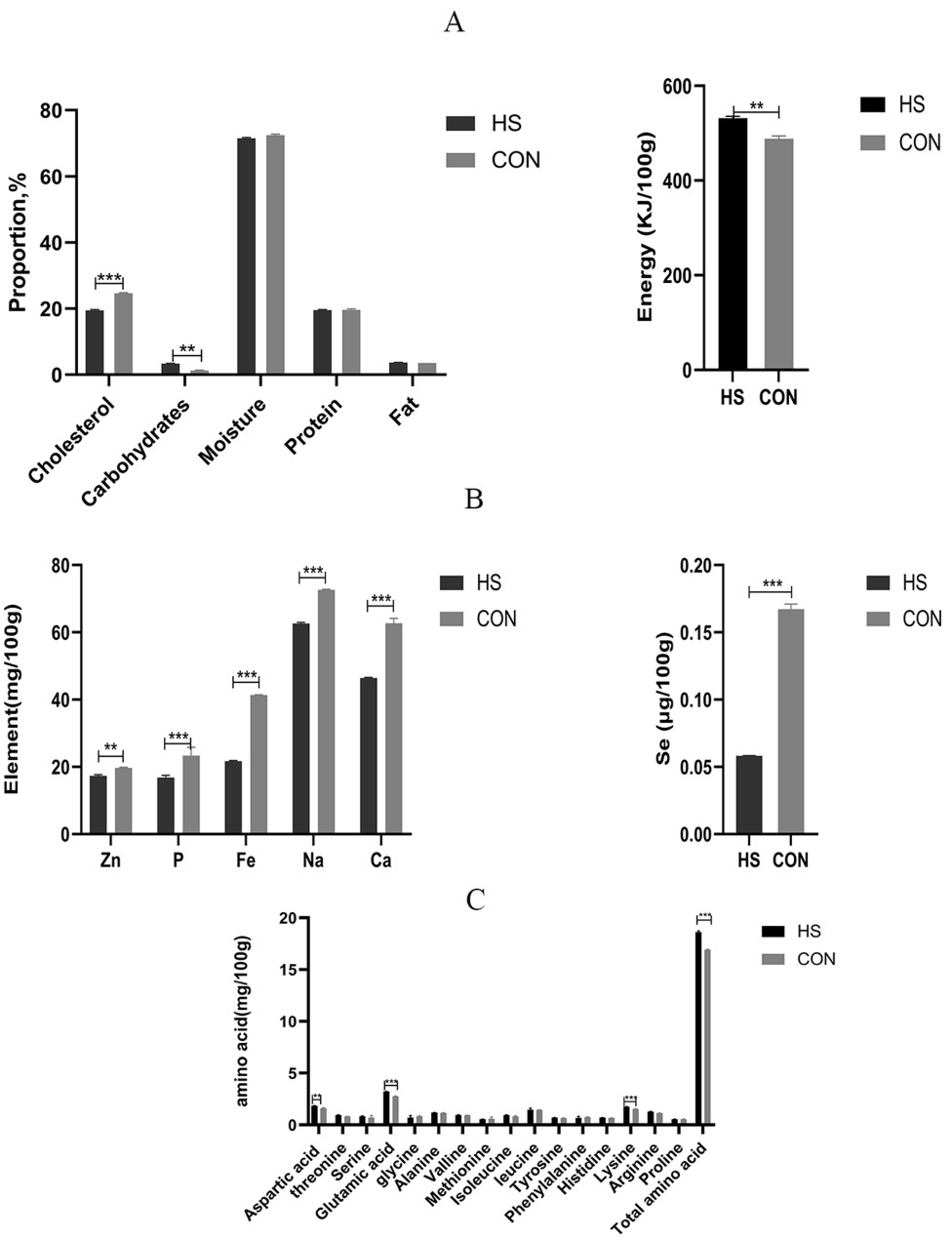

**Figure 4  Effect of herbal additives on the multi-component index of the Hungarian white goose muscle.** (A) The concentrations of carbohydrate, energy, moisture, cholesterol, protein, and fat in Hungarian white goose. (B) The concentrations of Zn, p, Fe, Na, Ca and Se in Hungarian white goose. (C) The concentrations of various amino acids in Hungarian white goose. *Significant difference compared to the Control group (*$P < 0.05$, **$P < 0.01$, ***$P < 0.001$, ****$P < 0.0001$). Data are the mean of 3 replicates of 2 samples each. All data was presented as mean values ± SEM, with appropriate standard error.

and *Fusobacterium* (6.61%) and a lower abundance of *Paraclostridium* (9.56%) than the HS group on day 70 (Fig. 6B). A heatmap was constructed using R language software to determine the relative abundance of the specific gut microbes. The results were illustrated

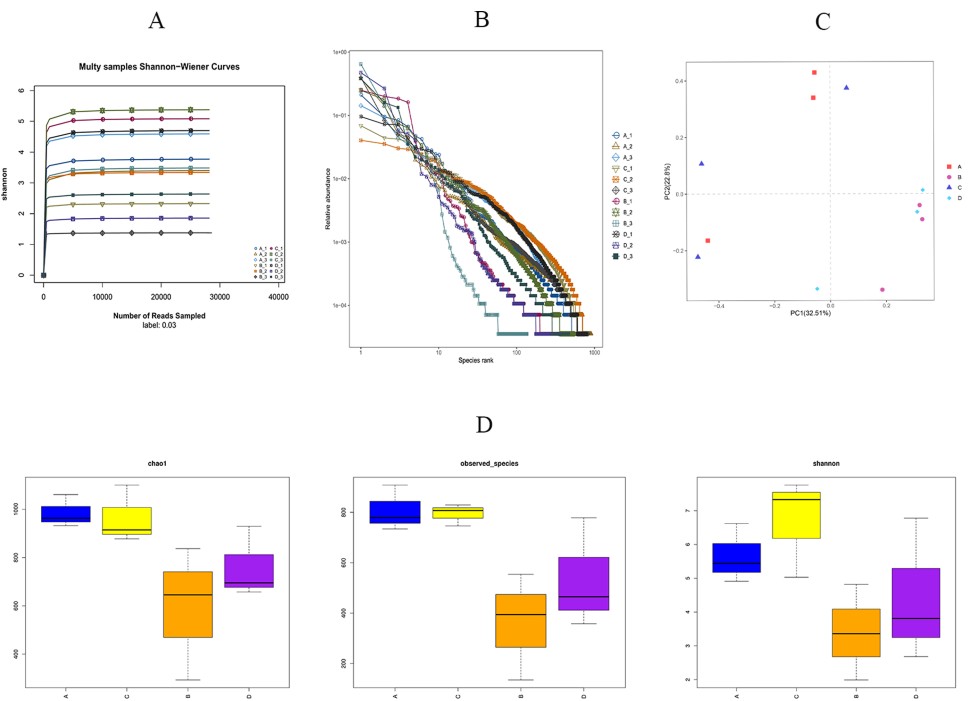

**Figure 5** **Effects of herbal mixture on the intestinal microbiota diversity in Hungarian white geese.** (A) Sample Shannon sparse curve. The horizontal coordinate indicates the number of sequences and the vertical coordinate indicates the Shannon value. When the curve tends to flatten, it indicates that the amount of sequencing data is large enough to reflect the vast majority of microbial information in the sample. (B) Rank Abundance Curve. The greater the abundance of species, the greater the range of the curve on the horizontal axis and the flatter the curve, the more evenly distributed the species. (C) Principal Coordinates Analysis (PCoA) based on weighted Unifrac metrics. (D) Alpha diversity analysis of four experimental groups. The horizontal coordinate represents the group name and the vertical coordinate represents the Alpha index. Chao1 and shannon were used as richness estimates. Observed-species index was used to indicate the number of OTUs observed with the increasing of the sequencing depth. A = HS group (P 42), B = HS group (P 70), C =Control group (P 42) and D = Control group (P 70).

in Fig. 7. The populations of Lactobacillus increased remarkably in the HS group on day 70. On day 70, *Escherichia-Shigella* and *Desulfovibrio* had lower abundance in the HS group compared with the CON group.

## Linear discriminant analysis effect size (LEfSe) analysis

A LEfSe analysis was performed to identify the specific bacterial taxa that may be responsible for the significant differences in community composition associated with CHAA and CHAB treatments. The cladograms revealed 41 potential microbial biomarkers with significant statistical differences. Among these, the abundance of three genera were significantly increased in the HS group than CON group on day 43, including *Lactobacillus sharpeae*, *Streptococcus_parauberis* and *Corynebacteriales*. *Epulopiscium*, *Lachnospiraceae bacterium mt14*, *Bacteria*, and *Paraclostridium* also showed significantly increased abundance in the HS group on day 70 compared with the CON group (Fig. 8).
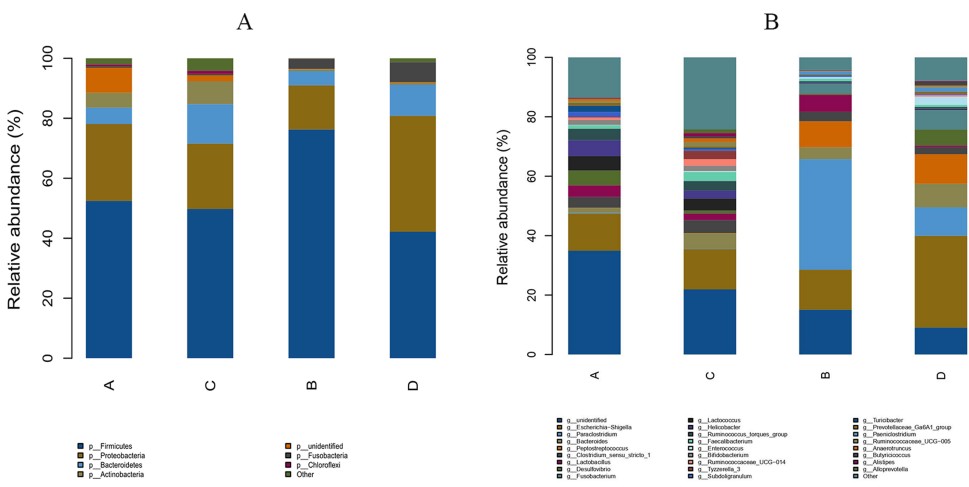

**Figure 6** **Effect of herbal additives on the composition of goose cecum flora at the phylum (A) and genus (B) levels.** Each bar represents the relative abundance of each bacterial taxon. A = HS group (P 42), B = HS group (P 70), C =Control group (P 42) and D = Control group (P 70).

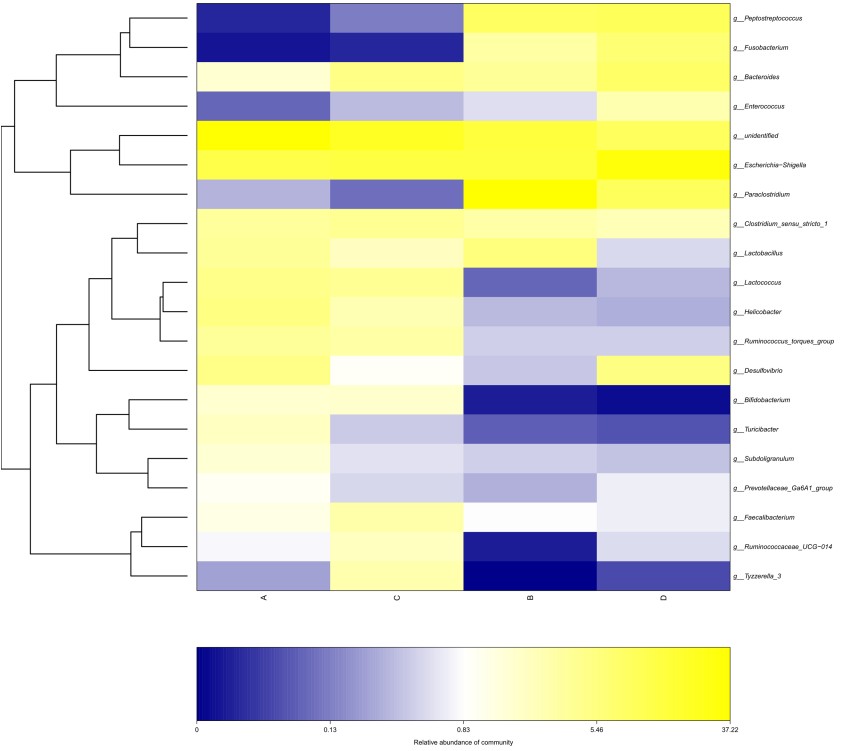

**Figure 7** **Heatmap depicting the relative abundance of each group of bacterial genera.** Group names are plotted on the $x$-axis and the $y$-axis represents each bacterial genus. A = HS group (P 42), B = HS group (P 70), C = Control group (P 42), D = Control group (P 70).

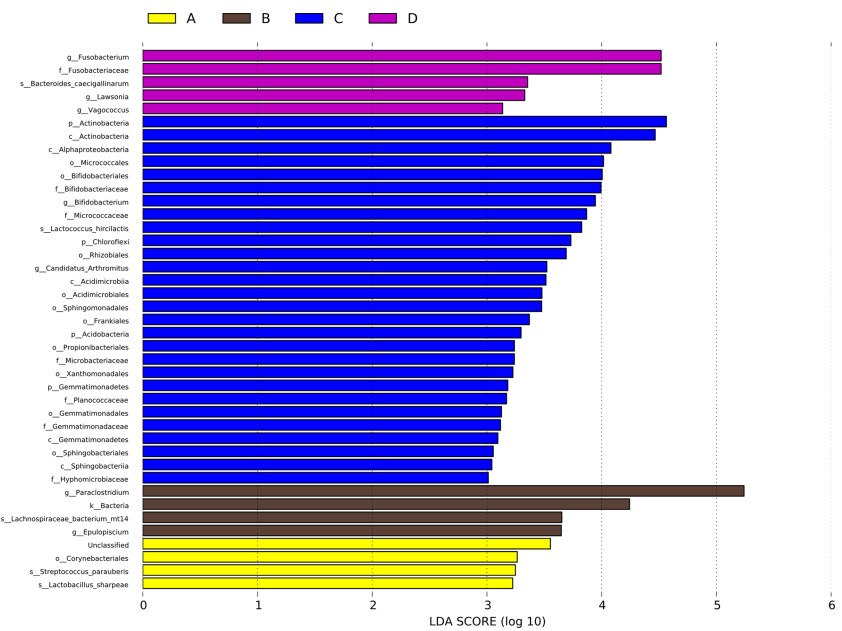

**Figure 8 LDA scores obtained from the LEfSe analysis of the gut microbiota in different groups.** Species with signifcantly diferent abundances in diferent groups are shown, and the length of the bar graph represents the efect size of the signifcantly diferent species. phylum to genus: p, phylum; c, class; o, order; f, family; g, genus. A = HS group (P 42), B = HS group (P 70), C = Control group (P 42), D = Control group (P 70).

## DISCUSSION

In recent years, the emergence of antibiotic resistance and antibiotic residues in food has become serious issues. Antibiotics are gradually banned as growth promoters in animal husbandry. Therefore, to develop antibiotic alternatives is of great interest and urgent for livestock industry. Herbal feed additives have become more common in the poultry industry nowadays (*Gao et al., 2022*). Several studies have revealed that combined herbal mixtures were more effective in improving animal health than single herbal extracts (*Xu et al., 2022*). However, the effect of Chinese herbal medicine mixtures on poultry production at different growth stages has not yet explored. This study aims to investigate the impact of CHAA, used at the starter phase and CHAB, used at the growth phase, on the meat quality, immunity and intestinal microbiota composition in Hungarian white geese.

Since the gosling has a short digestive tract, poor digestive gland function and weak immunity during the starter phase period (0-42), it is crucial to enhance digestion function and resistance against pathogens. CHAA contains *pulsatilla, gentian* and *Rhizoma coptidis,* which are widely known to have beneficial functions in antioxidant action, immune regulation, maintenance of the balance in intestinal microbiota, and the promotion of gastric juice secretion. Therefore, using CHAA can boost geese low resistance to pathogens and improve digestive system functions in the early stages (*Lyu et al., 2021*; *Zhang et al., 2021*; *Wang et al., 2018*; *Mirzae et al., 2017*). In the growth stage, digestion ability and immunity are improved and strong. However, bones, muscles, and feathers grow faster, it

is crucial to supply enough energy to enhance spleen and stomach functions. *Codonopsis pilosula*, *Atractylodes* and *Poria cocos* can regulate spleen and stomach function, which are demonstrated by the Qi-blood theory of Chinese medicine (*Gong & Hu, 2022*; *Cheng et al., 2009*; *Zhang et al., 2021*; *Yan et al., 2021*). As a result, CHAB was added to the basal diet during the grower phase (day 43 -70). Moreover, supplement of CHAA at 0.2% and CHAB at 0.15% to the basal diet promoted the growth performance of geese in our previous study. Therefore, we designed the experimental program (Fig. 1).

Immunoglobulins are essential indicators of the immune system and play an important role in the immune response (*Mikocziova, Greiff & Sollid, 2021*). Studies have shown that plant polysaccharides could improve the immune function of poultry (*Long et al., 2021*; *Yu et al., 2022*) and activate macrophages to exert immunomodulatory effects by recognizing and binding to specific receptors (*Zhang et al., 2021*). In our study, several herbs of CHAA and CHAB are rich in polysaccharides, such as *Codonopsis pilosula*, *Atractylodes* and *Poria cocos.* Besides, some other chemical components in herbal medicines also showed immune-enhanced effects, such as flavonoids in *Rhizoma coptidis* and glycyrrhizic acid in *Licorice* (Zhang, (*Liu et al., 2021*). Previous research showed that *Codonopsis pilosula*, *Poria cocos*, and *Licorice* supplementation could improve secretion of IgG, IgA and IgM (*Zhan, Yang & Xiao, 2015*; *Xie et al., 2013*; *Yin et al., 2022*). Our study found that supplementation with CHAA and CHAB enhanced humoral immune responses which was in agreement with these researches (Fig. 2).

PH value, filtration rate, shear force, fat content and flavor are indicators to evaluate meat quality traits. The accumulation of lactic acid caused by anaerobic respiration after slaughter will reduce meat's pH and affect meat water-holding capacity (*Ding et al., 2020*). The pH value of normal muscle was around 7, and decreased rapidly after slaughter (*Nkukwana et al., 2016*). In this study, there were no significant differences in the pH of goose meat among different treatments. This may be because these experimental geese were under similar management and nutritional conditions, including the isocaloric and isonitrogenous diet. Similarly, several studies have argued that Anacardium occidentale leaf and Moringa oleifera leaf supplementation did not influence on meat pH (*Adeyemi et al., 2021*; *Cui et al., 2018*). On the other hand, shear force and water loss rate are closely related to meat tenderness (*Guo et al., 2022*). Some studies have found that adding Astragalus and Glycyrrhiza complex polysaccharides could improve broiler meat quality by reducing muscle water loss and shear force (*Qiao et al., 2022*). In fact, CHAA and CHAB did not enhance the meat's tenderness, and there was no significant difference in HS group and the CON group in our study (Table 2). (*Yu et al., 2020*), also found that the supplementation of cottonseed meal to the feed had no effects on goose meat tenderness (*Yu et al., 2020*).

The composition and content of amino acids are important factors in meat quality and are frequently used to predict the nutritional value and flavor of meat (*Guo et al., 2022*). The composition and content of flavor amino acids, including Gly, Ala, Asp, Glu, Phe and Tyr, directly affect meat's freshness and flavor (*Liu et al., 2018*). Interestingly, the diet with CHAA and CHAB significantly increased the total amino acid levels, especially Glu, Lys and Asp levels. As a result, these findings proved that herb additives could improve the quality of goose meat by increasing the concentration of flavor amino acids and

total amino acids (Fig. 4C). Similarly, adding resveratrol increases the concentration of flavor-enhancing amino acid content in Pekin ducks, thereby improving the meat flavor. Cholesterol levels are also closely related to meat quality (*Omojola et al., 2015*). However, the increase in cholesterol content induces related diseases, such as cardiovascular diseases. In this study, cholesterol concentrations were significantly lower in geese fed with herb additives treatment than those in the CON group (Fig. 4A). In addition, studies have found that *Licorice* can exert anti-atherosclerotic effects by reducing cholesterol content (*Markina et al., 2022*; *Innih, Eze & Omage, 2022*). In our experiment, CHAB contains *Licorice*, which may explain the reduction of cholesterol content by supplementing CHAB. Furthermore, a certain amount of moisture contributes to the juiciness and tenderness of the meat and improve its quality. This study indicated that protein and moisture were not affected by the treatment of herbs additives. Intriguingly, we observed a higher muscle fat content in the HS group compared with that in the CON group (Fig. 4A). Fat content also could increase the juiciness and tenderness to improve the meat quality (*Yu et al., 2020*). Our data demonstrated that CHAA and CHAB increased the meat tenderness and flavor to some extent. The trace elements in the muscle of the HS group were significantly lower than those in the CON group (Fig. 4B). The exact mechanism will be needed to explore in the future.

Carcass traits are important indicators for evaluating animal production performance. Our results showed there were no difference in the percentage of tested carcass traits or weight for all treatments between the two groups (Fig. 3). These findings are consistent with previous studies (*Adeyemi et al., 2021*; *Yu et al., 2021*; *Miao et al., 2020*). This may be related to the fact that during avian development, carcass traits grow at a similar rate to body weight. The CHAA and CHAB used in our study did not increase the protein and energy content in the feed, resulting in similar carcass weights between the two groups.

Intestine is the primary site for food digestion and nutrient absorption, whose microbiota composition is important for homeostasis maintenance (*Zou et al., 2021*). In this study, we analyzed the microbiota of the caecum contents of Hungarian white geese by 16S rRNA sequencing. The results revealed that the predominant phyla were *Firmicutes*, followed by *Proteobacteria* and *Bacteroidetes* (Fig. 5, Fig. 6), which was in accordance with previous studies (*Li et al., 2017*; *Liu et al., 2018*). The *Firmicutes/Bacteroidetes* ratio (F/B) was significantly correlated with the capacity to obtain energy (*Kasai et al., 2015*). The present study found that dietary CHAA and CHAB supplementation increased this ratio. Moreover, *Lactobacillus*, as a probiotic, can improve poultry production performance and immunity by regulating gut microbiota (*Bian et al., 2016*). Our study showed that the addition of Chinese herbs increased the levels of *Lactobacillus*.

Furthermore, *Bacteroides* and *Desulfovibrio* have been reported to be important producers of LPS (*Diling et al., 2017*). At the genus level, we found that the abundance of *Bacteroides* and *Desulfovibrio* in the HS group decreased on day 70 (Fig. 7, Fig. 8), indicating that CHAB may inhibit the proliferation of these bacteria. *Escherichia-Shigella* is a pathogenic bacterium that causes diarrheal diseases, such as bacillary dysentery and hemorrhagic colitis (*Lee, Yoon & Tesh, 2020*). Studies have shown that flavonoids and saponins could inhibit the growth of pathogens such as *Escherichia* coli and *Pseudomonas*

(*Shu et al., 2020*; *Niu et al., 2023*). In addition, Polysaccharides inhibited the levels of pathogenic bacteria by promoting the production of short-chain fatty acids and organic acids in the intestine, significantly increasing the abundance of dominant bacteria (*Shi et al., 2020*; *Li, Zhao & Wang, 2009*). In our research, CHAA and CHAB decreased the abundance of *Escherichia-Shigella*. Taken together, CHAA and CHAB could shape the intestinal microbiota by increasing the abundance of beneficial bacteria and reducing the abundance of pathogenic bacteria.

## CONCLUSION

In conclusion, our data indicate that dietary supplementation of CHAA and CHAB significantly improved the meat quality and flavor by increasing the total amino acid content (Glu, Lys and Asp) and reducing the cholesterol content. Additionally, adding CHAA and CHAB could significantly stimulate and improve humoral immunity by increasing the levels of IgG, IgA and IgM in serum. CHAA and CHAB both showed an influence on increasing the beneficial microbiota abundance and decreasing the pathogenic microbiota abundance in the ceacum. Generally, these results provide valuable information that CHAA and CHAB could improve the meat quality, regulate immunity and shape the intestinal microbiota composition in Hungarian white geese. This study lays a foundation for developing and applying Chinese herbal medicine as a functional feed additive. The impact of the precise ingredient of Chinese herbal medicine on the growth performance and meat quality in geese will be investigated in the future.

## ACKNOWLEDGEMENTS

The authors thank the Goose Research Center of Jilin Agricultural University for providing the experimental animals, experimental platform used for this experiment and the related experimental equipment.

### Funding

This research was funded by the Jilin Province Science and Technology Development Project (20200201109JC, 20200301035RQ). The funders had no role in study design, data collection and analysis, decision to publish, or preparation of the manuscript.

### Grant Disclosures

The following grant information was disclosed by the authors:
Jilin Province Science and Technology Development Project: 20200201109JC, 20200301035RQ.

### Competing Interests

Chang Liu was employed by Changchun Animal Husbandry Service.

## Author Contributions

- Guilin Fu and Yuxuan Zhou conceived and designed the experiments, performed the experiments, authored or reviewed drafts of the article, and approved the final draft.
- Yupu Song, Chang Liu and Qiuyu Xie performed the experiments, analyzed the data, prepared figures and/or tables, and approved the final draft.
- Manjie Hu, Jingbo Wang, Yuxin Zhang, Yumeng Shi, Shuhao Chen performed the experiments, prepared figures and/or tables, and approved the final draft.
- Jingtao Hu and Yongfeng Sun conceived and designed the experiments, authored or reviewed drafts of the article, and approved the final draft.

## Animal Ethics

The following information was supplied relating to ethical approvals (i.e., approving body and any reference numbers):

All Hungary Geese were housed and used in accordance with the guidelines for animal experimentation set forth by the Animal Care and Use Committee of Jilin Agricultural University (Approval number. No. 2020 04 30 001. Date: 12 April 2020).

## Data Availability

The raw sequencing data generated from this study are available at NCB SRA: SRP409928.

## Supplemental Information

Supplemental information for this article can be found online at http://dx.doi.org/10.7717/peerj.15316#supplemental-information.

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
