# Peer review of "The effect of combined dietary supplementation of herbal additives on carcass traits, meat quality, immunity and cecal microbiota composition in Hungarian white geese"

_PeerJ, doi:10.7717/peerj.15316_

## Round 0.1 · original submission · Major Revisions

Please, reviewers have raised important points at various aspects of the current manuscript. The authors should carefully attend to all the concerns. Please authors should clearly justify the rationale for this current study. Also, authors should kindly provide recommendations for future studies at the end of the conclusion. Major revisions is needed given the totality of all the comments, which if diligently attended, would certainly elevate the quality of this work. Look forward to your revised manuscript.

·

Basic reporting

Discussion is need to be reorganized.

Experimental design

more detail is needed

Validity of the findings

the conclusion is too general.

Additional comments

General:
The background for the justification of A and B are missing.

Abstract:
Line 35-37: what is additive A and B?
Line 37: the experimental results show, delete “experimental”
Slaughter performance, meat quality (line 37-39) be specific for the parameters
Line 42: delete “in addition”
Line 49: “4” is for?
Line 49-50: this conclusion, avoid “maybe” and in general. Be specific what is the conclusion for the study according to the hypothesis
Keywords: use capital in every first word of each keywords

Introduction:
Line 54-55: personally, i disagree with this sentence. Improving the meat quality is not because people's quality of life improves. Improving meat quality is because it is needed to have healthy food
Line 84-102: before this paragraph, it would be useful to mention if those herbal plants are examples of Chinese medicinal herbal plants.

Materials and Methods:
Line 117: all animals were
Line 124-125: please add: “were prepared as follows.”
What is/are the reason/s for the different administration of A and B? Why A and B were not mixed, why A was administered first and B? Is there any scientific background for this administration
Line 141: why 0.2% A and 0.15% B?
Line 138-148: was this method adopted from other studies? Please include the references
Line 138-148: where were the geese obtained?
Please include all the references for the parameters evaluated
Line 171-174: why and so on? Please include all the elements studied.
I didnt find the analysis for cholesterol, carbohydrates, moisture, protein, fat in the methods. Also for energy, amino acids, alpha diversity analysis and beta diversity analysis.


Results
Fig. 1: I suggest removing the legends. It's already in the axis title
Line 205-207: according to the figure 2, only SR and BMR which had different results on HS and CON
Table 2: i could not find any sign (a-c) difference in the table
Line 209: please mention: the meat quality was evaluated based on shear force, etc….
Line 211: are those number for breast muscles or leg, or how this number was generated
Line 214: what are common elements?

Discussion:
The discussion is generally difficult to follow and to understand. I would suggest rewriting and to organise the discussion in order as it is in results (serum concentration, carcas traits, quality of meat and so on…)

Reviewer 2 ·

Basic reporting

The manuscript needs extensive English editing as many sentence aren't clear.

Relevant pieces of literature were cited

Experimental design

The experimental design is fine but the authors need to improve some sections of the methodology.

Validity of the findings

The results were not well written. Also, the conclusion needs some improvement to link the findings

Additional comments

Line 59 citation is needed
Line 67 define EU
Line 103 -111 please rephrase to improve understanding
Line 120 please add more details such as weight, how newborn healthy White hungary geese were obtained (did the authors hatch them or from different sources) etc this will enable repeatability of the work.
Line 126 state the room temperature
Line 126 powder (please state the size)
Line 133 to 136 please clarify…….do you mean you measure total saponin etc or?
Line 190 replicates are missing here
Line 199 seems like object………dive into your results
Overall the results section needs significant improvement
Please improve the conclusion
Fig 1c no stats on it?
Fig 2 no significant here ? why write “*Statistically signiûcant diûerence relative to the Control group (*P < 0.05, **P < 0.01, ***P <
0.001, ****P < 0.0001). SR=slaughter rate; HCR=half carcase rate; ER= eviscerated rate;
BMR= breast t muscle rate.”?
Figure 4 should be replace with better figures
Table 2 posthoc letters are missing

Reviewer 3 ·

Basic reporting

1. Your introduction needs to be detailed. I suggest that you improve the description at lines 84- 102. To provide more justifications for your study, you should expand upon the improvement of meat quality (tenderness, water holding capacity, etc.) by adding herbal additives to the diet.

2. In the discussion section, at lines 264-297, you justified the importance of six herbal mixtures used in the experiment. However, this information fits well in the introduction, therefore I suggest that you eliminate this portion and use some of it to reinforce the introduction.

3. The manuscript needs to be improved in phrasing, grammar, and vocabulary. Some examples where the language could be improved for clear understanding are 142 (it is either 6:00 am and 5:00 pm or 6:00 and 17:00), 299-300, 327, 340-342, 384-386, and 388-391.

4. China accounted for 94.1% of global geese production (Dibner & Richards, 2005) (line 61). This reference does not seem appropriate, kindly give a suitable reference.

5. Kindly support the statements displayed in the following lines with references: 131-134 and 359-362.

Experimental design

1. In the section materials and methods (lines 113-195), more citations from the scientific literature may be incorporated to support the validity and adequacy of the used methodologies.

2. Several studies including Ashour et al (Italian J of A. S, 2020, 19 no. 1, 1228-1237) have considered a concentration of 0.5 % of herbal additive in the diet. Explain why you used 0.2 and 0.15 (line 141).

3. Chinese herbs are dried, crushed, and pulverized into powder (125). Provide details on the drying process, for readers willing to reproduce the same exact experiment.

Validity of the findings

1. Figure 4 (a, b, c, and d), figure 5 (a, b, and c), and figure 6 (a and b) are not clear of vision. Please replace them with high-quality figures.

2. Some key results may be included in the conclusion section to support the conclusions statements.

Additional comments

1. Be consistent in italicizing names of species throughout the manuscript, for example, Astragalus, Escherichia coli, Salmonella, ..etc.

2. Breast and leg muscles were tested for 6 general indicators (172). What are those 6 indicators? Please list them in the manuscript.

---

## Round 0.2 · Minor Revisions

Please, authors kindly consider further revising your work given additional comments raised by the reviewers.

Grammar is critical to be thoroughly checked again. Please, do this carefully and meticulously.

Also, consider the following:

a) Please, start the materials and methods section with a new subsection captioned 'Schematic overview of the experimental program', comprising 3-4 sentences, and supported with a flow diagram. Please, use this tell us the major steps you followed to organise your study. Make sure that this directly connects and reiterates the objective of this work. The editor will be looking to examine this .

b) In your results, please provide the exact p-values, in addition to F-value, and R-sq values, all in bracket. I see you used SPSS and Graph Prism software, so this must be provided in all the places where p-value is mentioned in your results.

c) In your discussion, please provide more depth of the literature argument with the result. Tell us more about the how? and why? Do not just mention literature, but go further to contextualize, qualify and quantify your results with the literature. Also, please in the discussion, kindly ensure to indicate '(Refer to Figure x)' or '(Refer to Table x)' at all the places where specific results from them are mentioned. That is to say, all the figures and tables mentioned in the results must be mentioned in the discussion. I will carefully check this.

d) Make sure you clearly differentiate between your control , and other treatments, from the results as well as when it is mentioned in the discussion. Please, in the discussion, tell us why various tested parameters increased or decreased. There are some tiny discussion I see in the results section. Please, results should strictly be to relay the results. Kindly make effort to remove any kind of discussion in the results section. Results should simply be, this is what we see in the data. Discussion should be, this is why, and how this data is happening.

e) Please in your conclusion, kindly provide direction for future work, ok. Authors, please brainstorm on this, and come up with something ok

Look forward to your revised manuscript. Thank you

·

Basic reporting

No comment

Experimental design

No comment

Validity of the findings

No comment

Additional comments

General comments:
Please pay attention when to use space (“ ”). After “.” use space. Please check the whole manuscript.
Although the authors had mentioned that the manuscript has been checked by a native English speaker, I found that it's still necessary to re-check it again.

Abstract:
Line 45: no significance, i dont think its necessary to include

Introduction:
Line 62: …nutrition demands (Li et al. 2019)
Line 64: …including cardiovascular disease, …

Results:
Line 288-292: the results in table 2, the shear stress in HS and control: 102 vs 82, also 71 vs 59. As a reader, numerically, it's clearly different. But statistically, the reader can't see. Please confirm that the shear force had no significant effect between HS and control. Please include the letter which describes that they had no difference. Although posthoc test showed no significant difference, i suggest to add posthoc letter, use a same letter, for instance “a” in all numbers, showing that there is no difference.
Figure 5: its difficult to read the legend, is it possible to make the legend in fig.5 more clear?
Discussion:
Line 408-411: pH is one important factor, but the treatment actually did not influence the pH meat. What does it mean in this study?
Line 415-417: the same as previous question, if the treatment had no impact on geese meat quality, ? This sentence has a negative meaning. If I understand it well, the treatment generated the same pH, shear force, filtration rate, had the same level as in control. That means that the treatment generated the same quality parameter as in control. It does not mean that the treatment had no impact on meat quality. I would suggest presenting what is the standard for these parameters? This is important to emphasize the importance of the study. To avoid misunderstanding, I would suggest rewriting line 408-411 and line 415-417 as the style written in line 445-450. Although the content of protein and energy resulting is in control, the meaning of the sentence is positive.

Reviewer 2 ·

Basic reporting

The authors have revised and improved the manuscript according to the reviewers comments. Please accept the manuscript in the current form.

Experimental design

Good

Validity of the findings

Fine

Reviewer 3 ·

Basic reporting

The quality of the manuscript may be elevated by improving its grammar and vocabulary for clearer and more unambiguous understanding.

Experimental design

No comment

Validity of the findings

No comment

---

## Round 0.3 · accepted · Accept

The authors have carefully addressed all the comments raised by the reviewers. The editor has assessed the revised version and is very satisfied with the quality of the work. It is acceptable for publication. Thank you authors for finding PeerJ your journal of choice, and look forward to your future scholarly contributions. Congratulations and best wishes.